# Causal Association Between Psoriasis and Age-Related Macular Degeneration: A Two-Sample Mendelian Randomization Study

**DOI:** 10.3390/genes16121489

**Published:** 2025-12-12

**Authors:** Young Lee, Soojin Kim, Je Hyun Seo

**Affiliations:** 1Veterans Medical Research Institute, Veterans Health Service Medical Centre, Seoul 05368, Republic of Korea; lyou7688@gmail.com; 2Department of Ophthalmology, Veterans Health Service Medical Centre, Seoul 05368, Republic of Korea; soojin.sk@gmail.com; 3Department of Ophthalmology, Asan Medical Centre, University of Ulsan College of Medicine, Seoul 05505, Republic of Korea

**Keywords:** age-related macular degeneration, Mendelian randomisation, psoriasis, immune mechanism, single-nucleotide polymorphisms

## Abstract

Background/Objectives: Psoriasis and age-related macular degeneration (AMD) may share immune-related pathophysiologic characteristics. However, few studies have investigated the relationship between psoriasis and AMD. We assessed the possible causal link between psoriasis and AMD in European populations. Methods: Single-nucleotide polymorphisms associated with psoriasis exposure were employed as instrumental variables (IVs) based on genome-wide significance (*p* < 5.0 ×$\;10^{-8}$) in the FinnGen genome-wide association study (GWAS). The GWAS data for AMD were obtained from 11 studies performed by the International AMD Genomics Consortium. We performed a two-sample Mendelian randomisation (MR) study to estimate causal effects using the inverse-variance weighted, weighted median, and MR-Egger methods, as well as the MR-Pleiotropy Residual Sum and Outlier (MR-PRESSO) test. Results: We observed significant causal associations of psoriasis with AMD. Using the weighted median method, the odds ratio (OR) was 1.09 (95% CI = [1.03–1.16] and *p* = 0.005), and using the MR-PRESSO test, the OR was 1.04 (95% CI = [1.00–1.09] and *p* = 0.043). Conclusions: A potential causal association between psoriasis and AMD underscores the need to investigate inflammation as a risk factor for AMD.

## 1. Introduction

Psoriasis is a chronic, immune-driven condition affecting the skin characterised by hyperplasia and involves notable cutaneous changes, such as rashes, scaling, erythematous plaques, and itching [1]. It affects approximately 2–3% of the global population and is increasingly recognised as a systemic disease. Its associations extend beyond the skin to include metabolic syndrome, cardiovascular disease, and various autoimmune conditions. Age-related macular degeneration (AMD), on the other hand, is the most common cause of permanent vision impairment in the elderly worldwide [2,3,4], especially in those older than 60 years. It is a complex, multifactorial disease involving oxidative stress, chronic inflammation, genetic predisposition, lipid abnormalities, and smoking [2,5,6,7,8]. Both early and late stages of AMD have been associated with systemic inflammatory markers, suggesting that disrupted immune regulation plays a pivotal role in their development.

Several studies have reported potential immunological and molecular links between chronic inflammatory diseases such as psoriasis and neurodegenerative or vascular disorders. Given the systemic inflammatory nature of psoriasis and growing evidence concerning the involvement of inflammation in AMD development [9], psoriasis vulgaris may be associated with an increased risk of macular degeneration. Despite these theoretical connections, the relationship between psoriasis and AMD remains poorly understood. A recent study using data from the Taiwan National Health Insurance programme reported that patients with AMD had significantly higher odds of having psoriasis than controls (adjusted odds ratio (OR), 1.52; 95% confidence interval (CI), 1.07–2.15) [9]. Another recent study based on a U.S. national cohort reported an association between AMD and psoriasis with a risk ratio of 1.48 (95% CI: 1.37–1.60) after propensity score matching to adjust for potential confounders [10]. An analysis of the Korean National Health Insurance Service Claims Database revealed a significantly higher risk of neovascular AMD for patients with psoriasis than for controls (hazard ratio based on Cox proportional hazards models [11]: 1.33; 95% CI: 1.07–1.61).

Mendelian randomisation (MR) is an epidemiological method that utilises genetic variants as instrumental variables (IVs) to assess the causal impact of specific exposures they are linked to on disease outcomes [12,13]. MR studies often use existing genome-wide association study (GWAS) data, which makes them more cost-effective and time-efficient than randomised controlled trials. Several ophthalmological studies involving risk factor analysis using MR have been published recently [14,15,16,17]. However, there have been no study using the MR technique for association between psoriasis and AMD have suggested variable evidence. We aimed to investigate the potential causal effects of psoriasis on AMD through two-sample MR using summary statistics from FinnGen as the exposures and those from the International AMD Genomics Consortium (IAMDGC) as outcomes [18] and additionally performed multivariable MR adjusting for type 2 diabetes and hypertension.

## 2. Materials and Methods

### 2.1. Study Design

The institutional review board of the Veterans Health Service Medical Centre approved the study protocol (IRB No. 2023-03-005) and waived the requirement for informed consent due to the retrospective study design. The study was conducted in compliance with the tenets of the Helsinki Declaration.

### 2.2. Data Sources

Figure 1 illustrates the schematic overview of the analytical study design. The datasets used to explore the causal association between psoriasis and AMD were as follows. We used the FinnGen endpoint L12_PSORIASIS (“Psoriasis”), a broad psoriasis phenotype based on ICD-10 code L40, as the exposure dataset (*n* = 494,941; 12,760 cases of psoriasis and 482,181 controls; https://finngen.gitbook.io/documentation/data-download (accessed on 2 January 2025)), as shown in Table 1. For outcome dataset, we utilised the summary-level data derived from the 11 sources of IAMDGC GWAS data (*n* = 105,248; 14,034 cases and 91,214 controls) [19]. The FinnGen GWAS summary statistics for type 2 diabetes (*n* = 486,367; 82,878 cases and 403,489 controls) and hypertension (*n* = 478,149; 132,515 cases and 345,634 controls) were used for multivariable MR to assess potential confounding effects on the association between psoriasis and AMD. The datasets used for the summary statistics are listed in Table 1.

### 2.3. Determination of the Genetic Instrumental Variables

Single-nucleotide polymorphisms (SNPs) associated with each exposure at the GWAS threshold (*p* < 5.0 × $10^{-8}$) were used as IVs. These SNPs were pruned based on linkage disequilibrium (LD; r^2^ = 0.001, clumping distance = 10,000 kb) to ensure the independence of the IVs. The 1000 Genomes Phase III Dataset (European population) served as the reference panel for LD estimation during the clumping process. The F-value was calculated using the formula F = R^2^(n − 2)/(1 − R^2^), where n is the sample size and R^2^ is the proportion of exposure variance explained by genetic variance [20]. F-values of >10 indicated the absence of weak instrument bias [21]. The strength of the IVs was assessed using conditional F-statistics for the multivariable MR framework. A threshold of 10 was applied to denote sufficient instrument strength and minimise weak instrument bias [22].

### 2.4. Mendelian Randomisation

The MR analysis relied on the following assumptions regarding the IVs: (a) they had a strong association with the exposure, (b) they had no association with the confounders of the exposure-outcome relationship, and (c) they exerted effects exclusively on the outcomes through the exposure, indicating the absence of directional horizontal pleiotropy. We applied multiple MR approaches to estimate the causal association between psoriasis and AMD. These included inverse-variance weighted (IVW) MR with multiplicative random effects [21,23,24], weighted median estimation [25], MR-Egger regression with and without Simulation Extrapolation (SIMEX) correction [26,27], and MR-Pleiotropy Residual Sum and Outlier test (MR-PRESSO) [28]. The IVW method is most effective when all genetic variants satisfy the three assumptions for IVs [29]. However, the IVW estimates may be biased if one or more variants are invalid [25]. The weighted median approach produces accurate estimates of causality even if 50% of the instruments are incorrect [25]. The MR-Egger approach provides appropriate causal estimates even in the presence of pleiotropic bias. It permits a non-zero intercept, which reflects the average horizontal pleiotropic effects [26]. The MR-Egger with SIMEX can be used to correct the bias when the no measurement error assumption is violated, such as when the *I*^2^ value is <90% [27]. The MR–PRESSO test detects outlier variants and corrects the IVW estimates for horizontal pleiotropy by removing them [28]. Heterogeneity for IVW and MR-Egger was evaluated using Cochran’s Q and Rücker’s Q statistics, respectively [23,30]. The MR–PRESSO global test was used to assess directional horizontal pleiotropy. The results were interpreted based on the appropriate MR analysis method [31]. Possible pleiotropy in the genetic variants was indicated when Cochran’s Q, Rücker’s Q′, or MR–PRESSO global test yielded *p*-values < 0.05. Given that type 2 diabetes and hypertension are widely recognised risk factors for AMD, we conducted multivariable IVW MR analysis to estimate the direct effect of psoriasis on AMD while accounting for these potential confounders [22]. The Q_A_ statistic, which is a generalisation of Cochran’s Q, was used to evaluate heterogeneity among the instruments in the multivariable setting. We applied the Q-minimisation (Q-het) approach to obtain robust causal estimates in the presence of heterogeneity [22]. Standard errors were derived using a jackknife method [22]. Pairwise covariances for the SNP–exposure associations were required to compute conditional F and Q_A_ statistics for the exposures derived from overlapping samples [22]. These covariances were approximated using a phenotypic correlation matrix [22], which was estimated from the intercept of bivariate LD score regression [32,33,34]. The causal effect estimates are reported as ORs with 95% CIs. Statistical significance was set at *p* < 0.05 (two-sided). All analyses were executed using the TwoSampleMR, MVMR, MendelianRandomization, and simex packages (R version 3.6, R Core Team, Vienna, Austria).

## 3. Results

### 3.1. Genetic Instrumental Variables in Univariable Mendelian Randomisation

We identified 48 independent genetic variants associated with psoriasis based on genome-wide significance (*p* < 5.0 ×$10^{-8}$) and used them as IVs for the univariable MR analysis (Table 2). The mean F-statistic for these IVs was 507.95, and all individual F-statistics exceeded the conventional threshold of 10, indicating a low risk of weak instrument bias (Table 2 and Appendix A). Detailed information on the selected IVs is provided in Appendix A. Cochran’s Q test revealed significant heterogeneity among the instruments (*p* = 0.001; Table 2). Rücker’s Q′ test based on the MR-Egger method also indicated heterogeneity (*p* < 0.001). The MR-PRESSO global test suggested significant horizontal pleiotropic effects (*p* < 0.001). In contrast, the MR-Egger intercept test did not indicate directional pleiotropy both before (*p* = 0.854) and after SIMEX correction (*p* = 0.828). The Given the evidence of heterogeneity and pleiotropy, the MR-PRESSO results were regarded as the primary outcome. MR-PRESSO identified two outlier SNPs, prompting additional analyses in which heterogeneity and MR estimates were reassessed after outlier removal. These updated results are presented in Appendix A.

### 3.2. Univariable Mendelian Randomisation Analysis of the Causal Effects of Psoriasis on AMD

The univariable MR analyses identified significant causal links between genetically predicted psoriasis and AMD based on the weighted median (OR = 1.09, 95% CI: 1.03–1.16; *p* = 0.005) and the MR-PRESSO method (OR = 1.04, 95% CI: 1.00–1.09; *p* = 0.043). The IVW method revealed a marginally significant association (OR = 1.04; 95% CI: 0.99–1.09; *p* = 0.084) (Figure 2). The MR-Egger regression revealed a consistent direction of effect, although it was not statistically significant. This was evident from the odds ratios before and after SIMEX correction (OR = 1.05, 95% CI: 0.97–1.14; *p* = 0.259 and 0.253 before and after, respectively). The SNP-specific associations between psoriasis and AMD are demonstrated in the scatter plot (Figure 3). Following removal of the two MR-PRESSO outliers, heterogeneity was markedly reduced (Appendix A), and the updated IVW estimate became identical to the MR-PRESSO–corrected estimate. The weighted median re-mained statistically significant, and MR-Egger precision improved (*p* reduced from 0.25 to 0.08) while maintaining a consistent effect direction (Appendix A).

### 3.3. Multivariable Mendelian Randomisation Analysis of the Causal Associations Between Psoriasis and AMD

Multivariable MR analysis with adjustments for type 2 diabetes and hypertension revealed conditional F-statistics for psoriasis, type 2 diabetes, and hypertension of 13.09, 42.03, and 29.59, respectively. These indicated adequate instrument strength for each exposure (Table 3). The Q_A_ statistic was 427.31 (*p* < 0.001), indicating substantial heterogeneity among the IVs. We accounted for this heterogeneity by applying the IVW method with Q-minimisation (Q-het). The multivariable IVW (Q-het) analysis revealed a significant direct effect of psoriasis on AMD (OR = 1.056; 95% CI: 1.002–1.112) (Table 3). In contrast, type 2 diabetes (OR = 0.910; 95% CI: 0.794–1.044) and hypertension (OR = 1.020; 95% CI: 0.871–1.194) were not significantly associated with AMD (Table 3).

## 4. Discussion

Our research, based on two-sample MR analyses, provides evidence of a potential causal effect of psoriasis on AMD in European populations. We observed increased risk of AMD in patients with genetically predicted psoriasis based on 48 independently linked SNPs. The weighted median and MR-PRESSO methods revealed statistically significant associations. These results align with those reported in earlier epidemiological research from Taiwan, the United States, and Korea, which reported an increased risk of AMD among individuals with psoriasis [9,10,11]. 

The main strength of our study was the use of a relatively large cohort dataset, which enabled robust MR analysis and revealed a potential causal association between psoriasis and AMD. We minimised confounding and reverse causation, which are well-known limitations inherent to observational study designs, by leveraging summary statistics from large-scale GWAS consortia. The IVs used in our analysis were strong, as evidenced by their high F-statistics. Multiple MR approaches were also applied to account for pleiotropy and heterogeneity. Our study also adjusted for important comorbidities such as type 2 diabetes and hypertension, further supporting the independent effect of psoriasis on the risk of AMD.

The use of genetic instruments enabled us to explore the biological plausibility of the association and highlight shared inflammatory and immune-mediated pathways, including those involving the TNF-α, IL-17, VEGF, and complement system genes [35]. Converging evidence from immunological, genetic, and molecular studies supports a potential causal association between psoriasis and AMD. Both diseases were characterised by chronic inflammation and dysregulation of immune pathways. Psoriasis is accompanied by increased concentrations of inflammation- and angiogenesis-related mediators such as TNF-α, IL-17, and VEGF. These cytokines, which are central to skin inflammation, also play roles in retinal injury and pathological angiogenesis, which are hallmarks of AMD [36]. For example, TNF-α induces retinal ischaemia–reperfusion injury and axonal degeneration, and chronic elevation can lead to retinal pigment epithelium cell death and morphological changes resembling geographic atrophy, which is a late stage of AMD. TNF-α also regulates the expression of other inflammatory cytokines, including VEGF, and can modulate choroidal thickness and vascular integrity in the retina [37]. Experimental models have demonstrated the ability of IL-17 to stimulate angiogenesis in the retina independently of VEGF [38]. They have also demonstrated that complement activation further amplifies local inflammation and tissue damage. Observational and MR studies have reported that chronic systemic inflammation, as reflected by elevated C-reactive protein (CRP) concentrations, is associated with increased AMD risk. CRP can also contribute to local complement activation and retinal injury [16]. The complement system, a key component of innate immunity, is involved in both psoriasis and AMD. Genetic variants in complement pathway genes such as *CFH*, *C2*, and *C3* are established risk factors for AMD. Proteomic analyses of drusen from patients with AMD have revealed complement components (C5 and C9), CRP, and other inflammatory proteins. These findings indicate a pro-inflammatory environment in the macula.

Recent genetic studies, including the data presented in the Appendix A, support shared biological mechanisms between psoriasis and AMD. Several SNPs used as IVs in the psoriasis MR analyses are located in or near genes with known immunological and inflammatory functions relevant to both diseases, such as *IL23R*, *IL13*, *TRIM27*, *HLA-A*, *HLA-B*, and *TNFAIP3* [39,40].

*IL23R* (rs80174646) is a key regulator of Th17 cell differentiation and links psoriasis pathogenesis to retinal inflammation and neovascularization [41,42]. *IL13* (rs847), more commonly associated with Th2 responses, can modulate inflammation and tissue remodelling and influence retinal health. *TRIM27* (rs3132380) is involved in immune regulation and NF-κB signalling, pathways implicated in both skin and retinal inflammation [43].

*HLA-A* (rs9260057) is part of the MHC class I region and contributes to antigen presentation and immune surveillance in both skin and ocular tissues [44]. *HLA-B* (rs115860766) is strongly associated with psoriatic arthritis and immune-mediated diseases and may influence local immune responses in the retina [45]. *TNFAIP3* (rs622091) encodes A20, which is a negative regulator of NF-κB. Variants of this gene have been associated with excessive inflammation in psoriasis and possibly AMD. The presence of these SNPs among the IVs for psoriasis highlights the genetic overlap in immune regulation and inflammation between psoriasis and AMD. This supports the hypothesis that chronic immune activation—driven by cytokines, angiogenic factors, and complement dysregulation—contributes to the pathogenesis of both diseases and may underlie the increased risk of AMD in patients with psoriasis.

This study had some limitations. First, individual-level information could not be accessed. Therefore, we could not account for the confounding factors using summary statistics based on two-sample MR. Second, the tests used to assess the MR assumptions provide only partial rather than complete validation. Violations of the MR assumptions, such as horizontal pleiotropy or weak instrument bias, can lead to invalid conclusions and warrant prudence in interpreting the results. The initial MR analyses showed substantial heterogeneity, indicating that the MR assumptions were unlikely to hold. After the removal of two outlier variants identified by MR-PRESSO, heterogeneity was markedly reduced and was no longer statistically significant; however, the post–outlier removal *p*-value of 0.08 suggests that a small degree of residual heterogeneity cannot be completely excluded, and the findings should therefore be interpreted with caution. Third, in the multivariable MR analysis, the conditional F-statistic for psoriasis (F = 13.09) was borderline, raising the possibility of weak-instrument bias. While weak instruments in univariable two-sample MR generally bias estimates toward the null, weak instruments in two-sample MVMR may bias effect estimates either toward or away from the null depending on correlations among exposures and instruments. Thus, the MVMR findings should be considered supportive but not definitive. Fourth, the findings are based on populations of European ancestry and may not be generalisable to other ethnic groups. Finally, MR can strengthen causal inference, but it cannot fully account for complex gene–environment interactions or residual confounding. Further research involving individual-level data and functional studies is needed to confirm these findings.

## 5. Conclusions

Our findings provide strong evidence that psoriasis may increase the risk of AMD, likely through shared inflammatory and immune-mediated mechanisms. This study represents a significant advancement in the field by demonstrating the importance of systemic inflammation in the pathogenesis of AMD using rigorous MR methodology and a large-scale genetic dataset. These findings underscore the clinical relevance of monitoring ocular health in patients with psoriasis and highlight the need for interdisciplinary care. Further research, including functional studies and analyses involving diverse populations, is warranted to elucidate the underlying mechanisms and to confirm and extend these findings.

## Figures and Tables

**Figure 1 genes-16-01489-f001:**
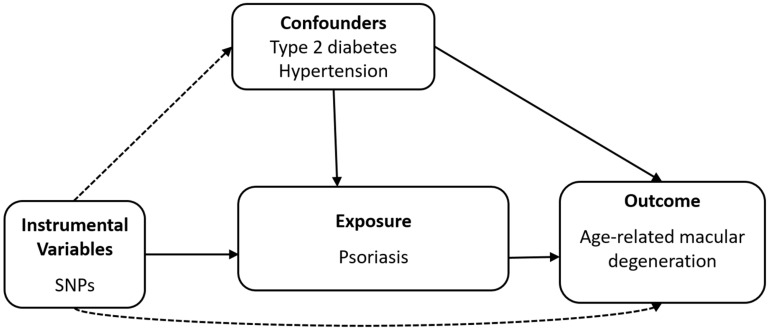
Diagram of two-sample Mendelian randomisation analysis. SNP, Single nucleotide polymorphism.

**Figure 2 genes-16-01489-f002:**
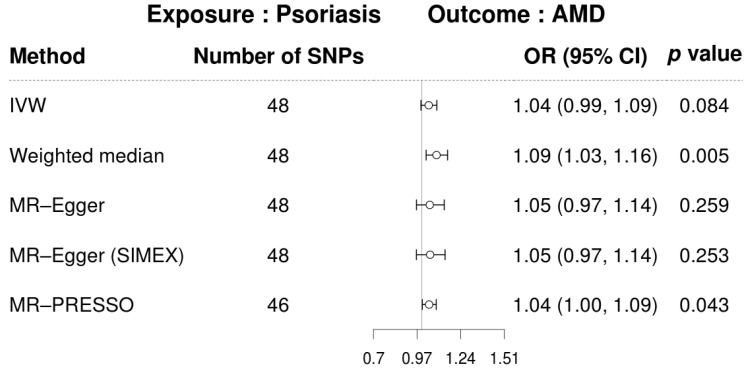
Forest plot of causal associations of psoriasis on AMD. AMD, age-related macular degeneration; CI, confidence interval; IVW, inverse-variance weighted; MR, Mendelian randomisation; OR, odds ratio; PRESSO, pleiotropy residual sum and outlier; SIMEX, simulation extrapolation; SNP, single-nucleotide polymorphism.

**Figure 3 genes-16-01489-f003:**
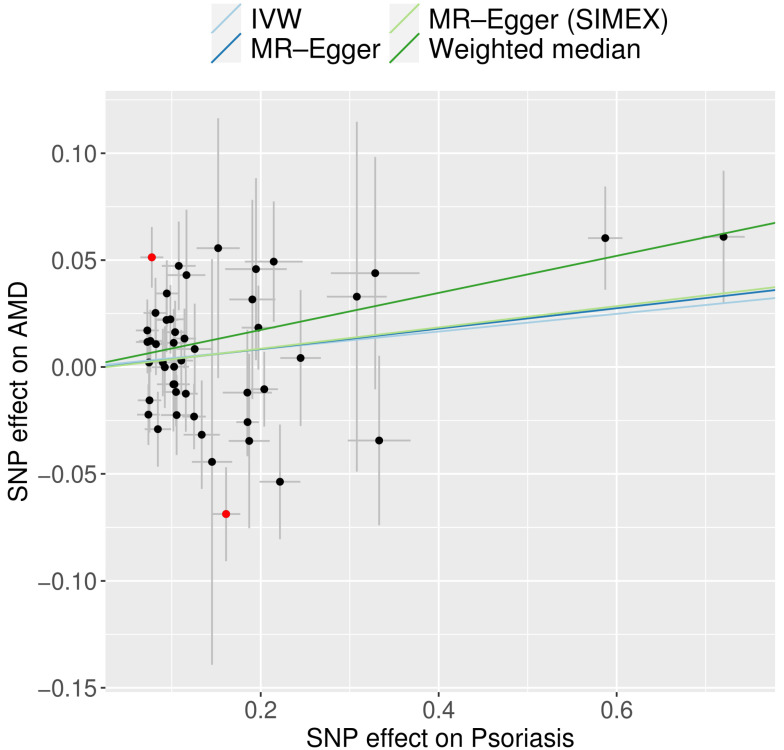
Scatter plot of MR tests assessing the effect of psoriasis on AMD. Dark blue, light blue, dark green, and light green regression lines represent the IVW, MR-Egger, MR-Egger (SIMEX), and weighted median estimates, respectively. The slope of the line represents the causal effect of each method. Each dot corresponds to a SNP, with the x-axis representing the association between the SNP and the exposure, and the y-axis representing the association between the SNP and the outcome. Red dots indicate outliers found in the MR-PRESSO analysis. AMD, age-related macular degeneration; IVW, inverse-variance weighted; MR, Mendelian randomisation; PRESSO, pleiotropy residual sum and outlier; SIMEX, simulation extrapolation; SNP, single-nucleotide polymorphism.

**Table 1 genes-16-01489-t001:** Summary statistics of data source.

Traits	Data Source	No. of Participants	Population	No. of Variants	Reference
Psoriasis	Finngen	494,941 (12,760 cases + 482,181 controls)	European	21,327,005	https://finngen.gitbook.io/documentation/data-download (accessed on 2 January 2025)
Type 2 diabetes	Finngen	486,367 (82,878 cases + 403,489 controls)	European	21,326,905	https://finngen.gitbook.io/documentation/data-download (accessed on 2 January 2025)
Hypertension	Finngen	478,149 (132,515 cases + 345,634 controls)	European	21,326,806	https://finngen.gitbook.io/documentation/data-download (accessed on 2 January 2025)
AMD	11 sources of data including the IAMDGC and UKB	105,248 (14,034 cases + 91,214 controls)	European	11,703,383	[19]

AMD, age-related macular degeneration; IAMDGC, International AMD Genomics Consortium; UKB, UK Biobank.

**Table 2 genes-16-01489-t002:** Heterogeneity and horizontal pleiotropy of instrumental variables.

Exposure				Heterogeneity		Horizontal Pleiotropy				
							MR-Egger		MR-Egger (SIMEX)	
	N	F	*I*^2^ (%)	*p* *	*p* #	*p* †	Intercept, β (SE)	*p*	Intercept, β (SE)	*p*
Psoriasis	48	507.95	97.18	0.001	<0.001	<0.001	−0.001 (0.006)	0.854	−0.001 (0.006)	0.828

* Cochran’s Q test from IVW; # Rücker’s Q’ test from MR-Egger; † MR-PRESSO global test. β, beta coefficient; F, mean F statistic; IVW, inverse-variance weighted; MR, Mendelian randomisation; N, number of instruments; PRESSO, pleiotropy sum of residuals and outlier; SE, standard error; SIMEX, simulation extrapolation.

**Table 3 genes-16-01489-t003:** Multivariable MR-IVW analysis with minimised Q statistic.

Exposures	Conditional F	Odds Ratio (95% CI)
Psoriasis	13.09	1.056 (1.002, 1.112)
Type 2 diabetes	42.03	0.910 (0.794, 1.044)
Hypertension	29.59	1.020 (0.871, 1.194)

CI, confidence interval; F, F statistic; IVW, inverse-variance weighted; MR, Mendelian randomisation.

## Data Availability

The datasets used and/or analysed in this study are available from FinnGen (https://finngen.gitbook.io/documentation/data-download, accessed on 2 January 2025) and the GWAS catalogue (https://www.ebi.ac.uk/gwas/summary-statistics, accessed on 1 December 2022).

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
