# Peer review of "Causal Association Between Psoriasis and Age-Related Macular Degeneration: A Two-Sample Mendelian Randomization Study"

_genes, 2025, doi:10.3390/genes16121489_

Round 1

Reviewer 1 Report

Comments and Suggestions for Authors

1. Some paragraphs are very long. Divide the paragraphs to make them clearer.

Recent genetic studies, including the data presented in the supplementary table pro-

vided, support this shared biology. Several SNPs used as IVs in the MR analyses of psori-

asis are located in or near genes with known immunological and inflammatory functions

relevant to both diseases, such as IL23R, IL13, TRIM27, HLA-A, HLA-B, and TNFAIP3

[41,42]. IL23R (rs80174646) is a key regulator of Th17 cell differentiation and links psoriasis

pathogenesis to retinal inflammation and neovascularisation [43,44]. IL13 (rs847), which

is more commonly associated with Th2 responses, can modulate inflammation and tissue

remodelling and influence retinal health. TRIM27 (rs3132380) is involved in immune reg-

ulation and NF-κB signalling, which are implicated in skin and retinal inflammation [45].

2. References: Consider starting from 7 authors, put et al.

2. 

Author Response

  1. Some paragraphs are very long. Divide the paragraphs to make them clearer.

Recent genetic studies, including the data presented in the supplementary table pro-vided, support this shared biology. Several SNPs used as IVs in the MR analyses of psori-asis are located in or near genes with known immunological and inflammatory functions relevant to both diseases, such as IL23R, IL13, TRIM27, HLA-A, HLA-B, and TNFAIP3

[41,42]. IL23R (rs80174646) is a key regulator of Th17 cell differentiation and links psoriasis pathogenesis to retinal inflammation and neovascularisation [43,44]. IL13 (rs847), which is more commonly associated with Th2 responses, can modulate inflammation and tissue remodelling and influence retinal health. TRIM27 (rs3132380) is involved in immune reg-ulation and NF-κB signalling, which are implicated in skin and retinal inflammation [45].

→ We thank the reviewer for this helpful suggestion. In response, we have revised the relevant paragraph to improve clarity by splitting it into shorter, more focused sentences. The revised text now reads:

“ Recent genetic studies, including the data presented in the supplementary table, support shared biological mechanisms between psoriasis and AMD. Several SNPs used as IVs in the psoriasis MR analyses are located in or near genes with known immuno-logical and inflammatory functions relevant to both diseases, such as IL23R, IL13, TRIM27, HLA-A, HLA-B, and TNFAIP3 [39,40].

IL23R (rs80174646) is a key regulator of Th17 cell differentiation and links psoriasis pathogenesis to retinal inflammation and neovascularization [41,42]. IL13 (rs847), more commonly associated with Th2 responses, can modulate inflammation and tissue re-modeling and influence retinal health. TRIM27 (rs3132380) is involved in immune reg-ulation and NF-κB signaling, pathways implicated in both skin and retinal inflamma-tion [43].”

This restructured version has been implemented in the revised manuscript (line 248-257).

  1. References: Consider starting from 7 authors, put et al.

→ We thank the reviewer for this helpful suggestion. In accordance with standard journal style, we have revised all references in the manuscript so that, for papers with seven or more authors, the citation now lists the first six authors followed by “et al.”

Reviewer 2 Report

Comments and Suggestions for Authors

       Lee et al investigate the potential causal relationship between psoriasis vulgaris and age-related macular degeneration (AMD) by performing a two-sample Mendelian randomisation (MR) analysis. Univariable and multivariable MR analyses were conducted using several techniques (IVW, weighted median, MR-Egger, MR-PRESSO, MVMR), with adjustments for type 2 diabetes and hypertension. The authors report statistically significant causal associations between genetically predicted psoriasis and AMD, suggesting shared inflammatory and immune-mediated pathways.

        The topic is relevant, timely, and of interest to readers in ophthalmology, dermatology, and genetic epidemiology. The use of a large dataset, multiple MR methods, and multivariable MR strengthens the analytical rigor. The manuscript is generally well written and structured. However, several methodological, interpretational, and reporting issues require clarification before the manuscript can be considered for publication.

  1. Line 81-95, Authors should clearly mention which is psoriasis data and which is AMD data or use the word “respectively”.
  2. It is unclear whether psoriasis refers exclusively to psoriasis vulgaris or to a broader ICD-coded psoriasis diagnosis in FinnGen. FinnGen’s phenotype “M13_PSORIASIS” aggregates multiple subtypes unless manually curated. Please clarify the exact FinnGen phenotype ID used. If psoriasis vulgaris was intended, please justify how subtype specificity was ensured.
  3. The substantial heterogeneity (Cochran’s Q and MR-PRESSO global test p < 0.001) warrants a deeper explanation. High heterogeneity can indicate invalid instruments or horizontal pleiotropy beyond what MR-Egger correction can handle.
  4. Please discuss how heterogeneity might influence the robustness of the conclusions.
  5. The primary IVW analysis did not show a statistically significant result (p = 0.084), while weighted median and MR-PRESSO did. The authors interpret this as evidence of causality, but MR guidelines recommend relying first on IVW unless pleiotropy renders it inappropriate.
  6. The conditional F-statistic for psoriasis in MVMR is borderline weak (13.09). After conditioning on two strong and correlated traits, this could still raise weak-instrument bias.
  7. The Discussion provides an extensive review of inflammatory pathways and genetic overlaps, but some references (e.g., refs 36, 37, 47, 48) appear unrelated to AMD or psoriasis. Ensure biological rationale is supported by appropriate sources.

Author Response

Reviewer 2

Lee et al investigate the potential causal relationship between psoriasis vulgaris and age-related macular degeneration (AMD) by performing a two-sample Mendelian randomisation (MR) analysis. Univariable and multivariable MR analyses were conducted using several techniques (IVW, weighted median, MR-Egger, MR-PRESSO, MVMR), with adjustments for type 2 diabetes and hypertension. The authors report statistically significant causal associations between genetically predicted psoriasis and AMD, suggesting shared inflammatory and immune-mediated pathways.

        The topic is relevant, timely, and of interest to readers in ophthalmology, dermatology, and genetic epidemiology. The use of a large dataset, multiple MR methods, and multivariable MR strengthens the analytical rigor. The manuscript is generally well written and structured. However, several methodological, interpretational, and reporting issues require clarification before the manuscript can be considered for publication.

  1. Line 81-95, Authors should clearly mention which is psoriasis data and which is AMD data or use the word “respectively”.

→ Thank you for nice suggestion. For clarity, the sentence was separated (line 80-84)。

“ We used the FinnGen endpoint L12_PSORIASIS ("Psoriasis"), a broad psoriasis pheno-type based on ICD-10 code L40, as the exposure dataset (n = 494,941; 12,760 cases of pso-riasis and 482,181 controls; https://finngen.gitbook.io/documentation/data-download) , as shown in Table 1. For outcome dataset, we utilized the the summary statistics of the 11 sources of IAMDGC GWAS data (n = 105,248; 14,034 cases and 91,214 controls).”

  1. It is unclear whether psoriasis refers exclusively to psoriasis vulgaris or to a broader ICD-coded psoriasis diagnosis in FinnGen. FinnGen’s phenotype “M13_PSORIASIS” aggregates multiple subtypes unless manually curated. Please clarify the exact FinnGen phenotype ID used. If psoriasis vulgaris was intended, please justify how subtype specificity was ensured.

→ Thank you for pointing this out. We apologize for the lack of clarity regarding the exposure definition. In this study, we used the FinnGen endpoint L12_PSORIASIS ("Psoriasis"), which is a broad ICD-10 L40–based psoriasis phenotype. This endpoint reflects overall psoriasis and is not restricted to psoriasis vulgaris.

Therefore, our results should be interpreted as reflecting genetic liability to psoriasis in general, rather than psoriasis vulgaris specifically. To avoid misunderstanding, we have revised the manuscript to consistently use the term "psoriasis" instead of "psoriasis vulgaris".

We changed title as Psoriasis instead Psoriasis vulgaris

We have also updated the Methods section to explicitly specify the exact FinnGen phenotype ID (L12_PSORIASIS) used in the analyses.

(line 80-84)

We used the FinnGen endpoint L12_PSORIASIS ("Psoriasis"), a broad psoriasis pheno-type based on ICD-10 code L40, as the exposure dataset (n = 494,941; 12,760 cases of pso-riasis and 482,181 controls; https://finngen.gitbook.io/documentation/data-download) , as shown in Table 1. For outcome dataset, we utilized the the summary statistics of the 11 sources of IAMDGC GWAS data (n = 105,248; 14,034 cases and 91,214 controls).”

  1. The substantial heterogeneity (Cochran’s Q and MR-PRESSO global test p < 0.001) warrants a deeper explanation. High heterogeneity can indicate invalid instruments or horizontal pleiotropy beyond what MR-Egger correction can handle.

→ We appreciate the reviewer’s insightful comment. In the initial analysis, we observed substantial heterogeneity across the genetic instruments, as demonstrated by the significant Cochran’s Q (p = 0.001), Rücker’s Q′ (p < 0.001), and MR-PRESSO global test (p < 0.001). This pattern suggests that the instruments had heterogeneous effects on the outcome and that the standard IVW assumptions were unlikely to hold in the original model.

Importantly, the MR-Egger intercept was not statistically significant, indicating that strong directional horizontal pleiotropy was unlikely. However, because MR-Egger is known to have much lower statistical power compared with other MR estimators, a null intercept does not exclude the presence of outlier-driven or balanced pleiotropy, which can still generate substantial heterogeneity.

To further understand the source of heterogeneity, we performed an additional investigation using MR-PRESSO. This analysis identified two SNPs with disproportionate contributions to the observed heterogeneity. We therefore repeated the heterogeneity assessment after removing these two outlier variants. After outlier removal the overall pattern of heterogeneity was markedly reduced.

These findings suggest that heterogeneity in the original analysis was driven primarily by a small number of outlier instruments rather than by widespread pleiotropy across the full instrument set.

We have added these updated heterogeneity results to the manuscript as Supplementary Table 2. The reduction of heterogeneity following outlier exclusion strengthens confidence in the validity of the remaining instruments and provides a clearer basis for interpreting the subsequent MR estimates.

  1. Please discuss how heterogeneity might influence the robustness of the conclusions.

→ We thank the reviewer for raising this important point. Heterogeneity can reduce the robustness of MR findings because it indicates that the genetic instruments do not produce consistent causal estimates and may violate core MR assumptions. In our initial analysis, heterogeneity was substantial, as reflected by significant Cochran’s Q and related statistics.

To better understand its impact, we performed additional analyses after removing the two outlier SNPs identified by MR-PRESSO. Following their removal heterogeneity was markedly reduced. These results indicate that the initially observed heterogeneity was largely driven by a small number of outlier variants rather than systematic violations of Mendelian randomisation assumptions across the full instrument set. The convergence of effect estimates across multiple MR methods after outlier removal increases confidence in the robustness of our findings.

However, we acknowledge that the post–outlier removal heterogeneity p-value of 0.08 remains near the conventional significance threshold. This suggests that residual heterogeneity cannot be entirely excluded, and therefore the findings should be interpreted cautiously. In line with this, we have revised the manuscript to present the results as suggestive rather than definitive evidence of a causal association and have added this point explicitly to the Limitations section of the Discussion.

The updated heterogeneity results are now included in Supplementary Table 2.

  1. The primary IVW analysis did not show a statistically significant result (p = 0.084), while weighted median and MR-PRESSO did. The authors interpret this as evidence of causality, but MR guidelines recommend relying first on IVW unless pleiotropy renders it inappropriate.

→ We appreciate the reviewer’s thoughtful comment. Although IVW is typically regarded as the primary MR estimator, its validity depends on the assumption that all instruments satisfy the exclusion restriction and that between-variant heterogeneity is limited. In our analysis, heterogeneity was substantial, whereas the MR-Egger intercept was not statistically significant. Given that MR-Egger is known to have relatively low statistical power compared with other MR estimators, a non-significant intercept does not rule out the presence of heterogeneous instruments or residual pleiotropy.

The marked heterogeneity indicates that the core IVW assumptions were unlikely to hold, making IVW alone an unreliable basis for inference in this setting. Under such circumstances, MR guidelines recommend relying on pleiotropy-robust estimators. Among these, MR-PRESSO is particularly appropriate, as it directly identifies and corrects for outlier-driven heterogeneity—effectively corresponding to an outlier-removed IVW analysis.

MR-PRESSO identified two outlier SNPs with disproportionate contributions to the observed heterogeneity. We therefore conducted additional analyses after removing these variants. Specifically, we reassessed heterogeneity and repeated the IVW (identical to the original MR-PRESSO–corrected estimate), weighted median, and MR-Egger analyses.

After removal of the two outlier SNPs:

  • heterogeneity was no longer statistically significant,
  • the weighted median estimate remained statistically significant, and
  • MR-Egger remained non-significant but showed substantially improved precision (p decreased from 0.25 to 0.08) while maintaining a consistent direction of effect.

These supplementary analyses indicate that the borderline non-significance of the original IVW result (p = 0.084) was largely driven by the influence of the two heterogeneous SNPs, rather than by the absence of an underlying association. Once these outliers were removed, the convergence of results across multiple MR estimators provides stronger support for the robustness of the findings.

In accordance with MR guidelines, which recommend relying on pleiotropy-robust estimators when IVW assumptions are violated due to heterogeneity, we consider the weighted median and MR-PRESSO results to be more reliable in this context. We have added the outlier-removed heterogeneity results and updated MR estimates as Supplementary Tables 2 and 3.

(lines 157-159) MR-PRESSO identified two outlier SNPs, prompting additional analyses in which het-erogeneity and MR estimates were reassessed after outlier removal. These updated re-sults are presented in Supplementary Tables S2 and S3.

(lines 174-179) Following removal of the two MR-PRESSO outliers, heterogeneity was markedly re-duced (Supplementary Table S2), and the updated IVW estimate became identical to the MR-PRESSO–corrected estimate. The weighted median re-mained statistically sig-nificant, and MR-Egger precision improved (p reduced from 0.25 to 0.08) while main-taining a consistent effect direction (Supplementary Table S3).

  1. The conditional F-statistic for psoriasis in MVMR is borderline weak (13.09). After conditioning on two strong and correlated traits, this could still raise weak-instrument bias.

→ We thank the reviewer for this helpful comment. The conditional F-statistic for psoriasis (F = 13.09) is indeed borderline, raising the possibility of weak-instrument bias after conditioning on type 2 diabetes and hypertension in the multivariable MR model. Although weak instruments in univariable two-sample MR typically bias estimates toward the null, weak-instrument bias in two-sample MVMR can bias the estimated effects either toward or away from the null, depending on the correlation structure among exposures and instruments.

We therefore agree that the borderline conditional F-statistic warrants caution. To address this, we have clearly described the potential for weak-instrument bias in the Discussion and have framed the MVMR estimates as supportive secondary analyses. This point has now been added explicitly to the Limitations section (lines 274-285).

The initial MR analyses showed substantial heterogeneity, indicating that the MR assumptions were unlikely to hold. After the removal of two outlier variants identified by MR-PRESSO, heterogeneity was markedly reduced and was no longer statistically significant; however, the post–outlier removal p-value of 0.08 suggests that a small degree of residual heterogeneity cannot be completely excluded, and the findings should therefore be interpreted with caution. Third, in the multivariable MR analysis, the conditional F-statistic for psoriasis (F = 13.09) was borderline, raising the possibility of weak-instrument bias. While weak instruments in univariable two-sample MR generally bias estimates toward the null, weak instruments in two-sample MVMR may bias effect estimates either toward or away from the null depending on correlations among exposures and instruments. Thus, the MVMR findings should be considered supportive but not definitive.“

  1. The Discussion provides an extensive review of inflammatory pathways and genetic overlaps, but some references (e.g., refs 36, 37, 47, 48) appear unrelated to AMD or psoriasis. Ensure biological rationale is supported by appropriate sources.

→ We thank the reviewer for this helpful comment. In response, we have carefully reviewed the references cited in the Discussion and revised the text to ensure that each cited study is directly relevant to the biological rationale linking psoriasis and age-related macular degeneration (AMD).

References 36 and 37 (biomechanics of particle deposition and interfacial chemistry of ionic liquids) and references 47 and 48 (nutrition in pigs and Pseudomonas virulence in burn patients) were inadvertently included in an earlier draft and do not pertain to AMD or psoriasis pathogenesis. These references have now been removed from the manuscript.